# Slab morphology and deformation beneath Izu-Bonin

Haijiang Zhang[1,2], Fan Wang [ID] [1,4], Robert Myhill[3] & Hao Guo[1]

Seismic tomography provides unique constraints on the morphology, the deformation, and (indirectly) the rheology of subducting slabs. We use teleseismic double-difference P-wave tomography to image with unprecedented clarity the structural complexity of the Izu-Bonin slab. We resolve a tear in the slab in the mantle transition zone (MTZ) between 26.5° N and 28° N. North of the tear, the slab is folded in the MTZ. Immediately above the fold hinge, a zone of reduced P-wavespeed may result from viscous dissipation within an incipient shear zone. To the south of the tear, the slab overturns and lies flat at the base of the MTZ. The ~680 km deep 2015 Bonin earthquake (Mw~7.9) is located at the northernmost edge of the overturning part of the slab. The localised tearing, shearing and buckling of the Izu-Bonin slab indicates that it remains highly viscous throughout the upper mantle and transition zone.

[1] School of Earth and Space Sciences; Laboratory of Seismology and Physics of Earth's Interior, University of Science and Technology of China, 96 Jinzhai Road, Hefei 230026 Anhui, China. [2] CAS Center for Excellence in Comparative Planetology, 96 Jinzhai Road, Hefei 230026 Anhui, China. [3] School of Earth Sciences, University of Bristol, Queen's Road, Bristol BS8 1RJ, UK. [4]Present address: Department of Earth and Environmental Sciences, Michigan State University, Natural Science Building, East Lansing, Michigan 48824, USA. Correspondence and requests for materials should be addressed to H.Z. (email: zhang11@ustc.edu.cn) or to R.M. (email: bob.myhill@bristol.ac.uk)

The subduction of the oceanic lithosphere into Earth's convecting mantle is typically associated with narrow sub-planar regions of earthquakes called Wadati–Benioff zones. These earthquakes can be as deep as 650–700 km[1], providing strong evidence that subducting slabs penetrate into the mantle transition zone (MTZ). Since the early 1990s, independent constraints on slab extent and morphology have been provided by seismic tomography, which has imaged high seismic wavespeed anomalies associated with regions of subduction[2–4]. Tomographic studies have demonstrated that contiguous slabs can extend well beyond the limits of deep seismicity. Some of these slabs penetrate aseismically into the lower mantle, while others stagnate in the MTZ[3,5–8]. Slabs exhibit a wide range of morphologies, from near-planar (e.g., Japan–Kamchatka) to folded and contorted (e.g., Tonga–Fiji–Kermadec) and from contiguous to torn or thinned[9–12]. These variations in slab morphology must be a function of the history and distribution of forces acting on the slab, and the material properties of the slab itself. In this study, we seek to constrain the shape of the Izu–Bonin slab using high-resolution tomographic images, and use the morphology to make inferences about the physical characteristics and geodynamics of the region.

The Izu–Bonin subduction zone (Fig. 1) is a thousand-kilometre long system, which extends south from Tokyo, where the Pacific Plate subducts beneath the Philippine Sea Plate. At the northern end of the zone, the Izu–Bonin Trench meets the Japan Trench and the Sagami Trench at the Boso Triple Junction. At the southern end (east of the Volcano Islands), the strike of the Izu–Bonin trench rotates from north–south to northwest–southeast, marking the start of the Marianas Trench.

After subduction at the trench, the Izu–Bonin slab descends westward into the upper mantle and transition zone. At 100–400 km depth, the distribution of earthquakes indicates that the dip of the slab increases from 40° in the north to 80° in the south[13]. This increase in slab dip has been attributed to a higher velocity of trench advance in the south relative to the north over the last few million years[14–17]. At greater depths, earthquake locations indicate that the dip of the slab decreases to 20–30° along a fold hinge that deepens towards the south from ~400 km depth at 33°N to ~550 km depth at 27°N[18]. Geodynamic modelling suggests that this morphology may be attributable to westward subduction of the Philippine Sea Plate at the Ryukyu Trench, 2000–3000 km west of the Izu–Bonin slab[19–21]. Subduction at the Ryukyu trench induces a slab pull on the shallow part of the Izu–Bonin slab, and a positive dynamic pressure between the two subduction systems in the upper mantle and the transition zone. These should encourage the Izu–Bonin slab to steepen at intermediate depth and become increasingly convex in the direction of subduction in the upper mantle. In the MTZ, some simulations indicate buckling of the slab, while others indicate overturn[19,21], such that the tip of the slab can either face in, or opposite to, the direction of subduction at the trench.

Although earthquake hypocentres provide good estimates of slab location wherever they occur, some parts of the slab are aseismic. Other techniques are required to determine the extent and orientation of the slab in the MTZ, the nature of slab deformation and whether the slab directly penetrates into the lower mantle. High-resolution seismic tomography should be able to answer these questions, but existing studies have produced differing tomographic models and thus different interpretations of the deep slab. For example, tomographic and receiver function images of the southern end of the Izu–Bonin slab have variously been interpreted as evidence for direct penetration into the lower mantle[22–25], multiple isoclinal folds[26] or "heel"-shape thickening[27]. These differing interpretations imply different slab rheologies; for example, folding without thickening implies that the slab is much more viscous than the surrounding mantle, while significant slab thickening indicates a smaller viscosity contrast[28].

High-resolution seismic tomography could also help resolve the mystery of a very enigmatic deep earthquake. The May 30, 2015 Bonin Islands earthquake had a hypocentral depth of ~680 km, one of the deepest in the historical record[29]. More significantly, it ruptured ~100 km beneath and to the east of the Izu–Bonin Wadati–Benioff zone. Existing attempts to locate the earthquake relative to the slab disagree on the slab morphology, but all argue that it occurred close to the bottom of a 100–200 km thick slab[25–27,30]. Such a location would probably be both hot and dry, as water cannot penetrate so deep into the slab, either before[31,32] or after[33] subduction. This combination of high temperatures and dry conditions is a problem for earthquake-generating mechanisms involving thermally activated shear instabilities[34], transformational faulting[35] or dehydration embrittlement[36], all of which require low temperatures in the region of earthquake nucleation (probably not exceeding 800 °C).

Here, we obtain high-resolution tomographic images for the Izu–Bonin slab from teleseismic double-difference (DD) tomographic inversions[37,38] (see the Methods section) and present a structural framework that helps understand its deformation, its relationship to the tectonic history of the area and the anomalous nature of deep-focus seismicity. We resolve a tear in the slab, which can be approximated by a northward-dipping plane that splits the slab in the MTZ between 26.5° N and 28° N (500–670 km depth). North of the tear, the slab is folded in the MTZ, with the lower limb dipping shallowly to the west. We also image a zone of reduced P-wavespeed (Vp) at ~420 km depth in the slab above the fold hinge, which is apparently due to viscous dissipation within an incipient shear zone. To the south of the tear, the slab is completely overturned and lies inverted on the bottom of the MTZ. The imaged morphology of the Izu–Bonin slab indicates that it has a strongly non-Newtonian rheology throughout the upper mantle and transition zone. The 2015 Bonin earthquake occurred at the northernmost edge of the overturning slab, ~20–30 km from the slab–mantle wedge interface. Relatively low temperatures, high thermal gradients and high stresses may all have facilitated this large isolated earthquake.

## Results

**Seismic tomographic images for the Izu–Bonin slab.** In contrast to conventional tomographic techniques, which rely heavily on a good distribution of regional stations not present above most oceanic subduction zone systems, teleseismic DD tomography minimises the misfit between observed and theoretical travel times between pairs of earthquakes by simultaneously relocating those earthquakes and adjusting local seismic wavespeeds. Because the ray paths from pairs of events mostly overlap outside the source region, DD tomography is particularly sensitive to the wavespeed structure close to seismically active areas. As a result, it produces clearer images of subduction zones throughout the upper mantle and yields more accurate relative earthquake locations. It is particularly suited to the Izu–Bonin subduction zone, which has a good distribution of seismicity from the surface to >500 km depth. If differential arrival times for event pairs are not constructed by the waveform cross-correlation technique, however, they may contain larger random errors due to differencing in absolute arrival times. Thus, outliers in arrival-time picks should be carefully rejected.

We assembled arrival-time data for earthquakes occurring in the Izu–Bonin region for the period of 1960–2008 from the Engdahl–van der Hilst–Buland (EHB) catalogue located by the single-event method of Engdahl et al.[39]. A DD tomographic

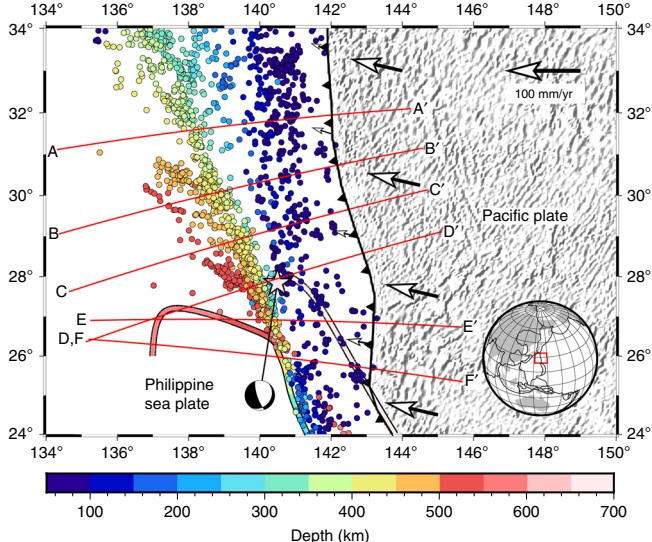

**Fig. 1** Overview map of the Izu–Bonin study area. The illumination shows the relief of the Pacific Plate (derived from Global Digital Elevation Model ETOPO2) to the east of the trench (Bird[59]). The convergence velocity between the Pacific and Philippine Sea Plates, and the rate of trench advance relative to the no-net reference frame of Argus et al.[60] are shown as white arrows. Relocated earthquakes from the EHB catalogue (1960–2008) are coloured according to hypocentral depth. The same colour scheme is used to indicate the depth of the edges of the highest velocity parts of the slab, as estimated from the tomographic inversion. The divergence at around 140.5E, 26.5 N marks the shallow limit of the tomographically observed tear. Lines A–A' through F–F' correspond to the tomographic sections (**a-f**) plotted in Fig. 2. Note that along Sections D-D' and E-E', the deepest parts of the slab extend both towards the west and the east; this is a consequence of the orientation of the slab tear, which is not perpendicular to the trench. The location of the 2015 Bonin Islands deep-focus earthquake is plotted as a star

model was constructed for Vp beneath the Izu–Bonin region, for which several cross sections are shown in Fig. 2. The spatial resolution is on the order of 30–40 km in longitude and depth and 100–200 km in latitude. Resolution tests show that the slab is robustly resolved by the data, but wavespeed amplitudes may be underestimated due to smoothing and damping regularisations applied to the DD tomographic system (see Supplementary Information). Our DD tomographic inversion also simultaneously relocates the 2015 Mw 7.9 Bonin earthquake at latitude 27.741°N, longitude 140.572°E and depth 679.9 km, which is deeper than the location (latitude 27.740°N, longitude 140.590°E and depth 667.2 km) given by Zhao et al.[25], but with a similar epicentre. The restoration test shows that our location uncertainty in depth is about 1.4 km (see Supplementary Information).

In the north of the area (around 31.5°N; Fig. 2a), a continuous high Vp anomaly dips with an angle of ~50° towards the west at intermediate depths (100–400 km). The Wadati–Benioff zone in this region is ~20 km thick, and is located close to the maximum wavespeed anomaly. At about 400 km depth, the dip of the high-wavespeed anomaly decreases to 20–30°. Further south (Fig. 2b, c), the dip of the high-wavespeed anomaly increases at intermediate depths and decreases in the MTZ, in agreement with previous studies[13]. At 28°N, the high-wavespeed anomaly abruptly bends through 80° to lie almost horizontally in the MTZ. The deepest relocated outboard earthquakes along this section have depths of about 540 km. At about 27°N (Fig. 2d), the flat-lying high-wavespeed layer becomes less prominent at the western end of the

section, and a stubby high-wavespeed anomaly appears below the bend in the Wadati–Benioff zone. In Fig. 2e, the flat-lying anomaly to the west becomes even shorter, and the wavespeed anomaly to the east becomes longer and merges with the shallower anomaly. Finally, in Fig. 2f, a high-wavespeed anomaly dips almost vertically from 200- to 500 km depth and then overturns towards the east to lie flat at the base of the MTZ.

## Discussion

Although the slab appears to be stagnant in the MTZ in the study region, it is possible that a small fraction of the slab in the north of the study region may penetrate through the 660-km interface, as indicated by relatively well-resolved high-wavespeed anomalies below 660 km in sections AA' and BB' (Fig. 2). To the east of the main Wadati–Benioff zone, there exists a high-wavespeed zone at depths from 0 to 400 km that dips away from the slab. The resolution tests (see Supplementary Information) show that at depths from 0 to 200–400 km around the eastern edge of the study region where the high-wavespeed anomaly is located, the model is not well resolved due to poor angular coverage of ray paths. Therefore, it is likely that the high-wavespeed anomaly at the edge of the model is not robust, and no further interpretation is attempted here.

We interpret the high-wavespeed anomalies in Fig. 2 as a single contiguous slab undergoing tearing. In Sections AA'–CC', the locations of the high-wavespeed anomalies are in excellent agreement with previous interpretations based on seismicity and tomography. This good agreement is probably due to the relatively simple shape of the slab in the north of the study region, and the proximity to the dense Japanese seismic network. At the southern end of the study area, the improved resolution of the DD tomography method yields the first clear image of the morphology of the slab. The abrupt change in orientation of the high-wavespeed anomalies in the MTZ seen in Sections CC'–EE' requires the presence of a narrow slab tear at least 500 km long. Our interpretation is shown in Fig. 3. The northern edge of the tear marks the southern termination of the sub-horizontal slab, while the southern edge of the tear dips at an angle of ~45° to the NNE, beneath the bend in the main Wadati–Benioff zone. The tear decouples the northern and southern parts of the deep slab, allowing the southern end to overturn, making Izu–Bonin the first reported subduction zone where the slab lies inverted on top of the upper–lower mantle boundary. The overturned slab fragment in Fig. 2f appears to be about 300 km long. We propose that the relatively low wavespeeds observed over a similar distance further west in the same cross section correspond to the slab gap created during tearing (Fig. 3). A similar interpretation was made by Zhao et al.[25].

The slab morphology observed in our tomographic images closely resembles the morphologies produced by geodynamic simulations, which model two closely spaced (<3000-km) slabs subducting with the same polarity[19–21], like the Izu–Bonin and Philippines slabs. These simulations impose similar plate velocities to those seen in the Izu–Bonin–Philippines region, but do not use output from tomographic inversions as input to the simulations. Over time, the simulations show the equivalent of the Izu–Bonin Trench advancing, with the slab increasing in dip and either flattening[20,21] or becoming overturned[19] in the MTZ. The time evolution in the 2D geodynamic models can be used as an approximation for the spatial evolution southwards along the arc, as the cumulative distance of trench advance is greater at the southern end of the Izu–Bonin system.

The magnitude of the high-wavespeed anomaly to the north of the slab tear decreases by ~1% above the tight bend in the slab (Sections BB', CC', DD'), but not further north (Section AA'). A

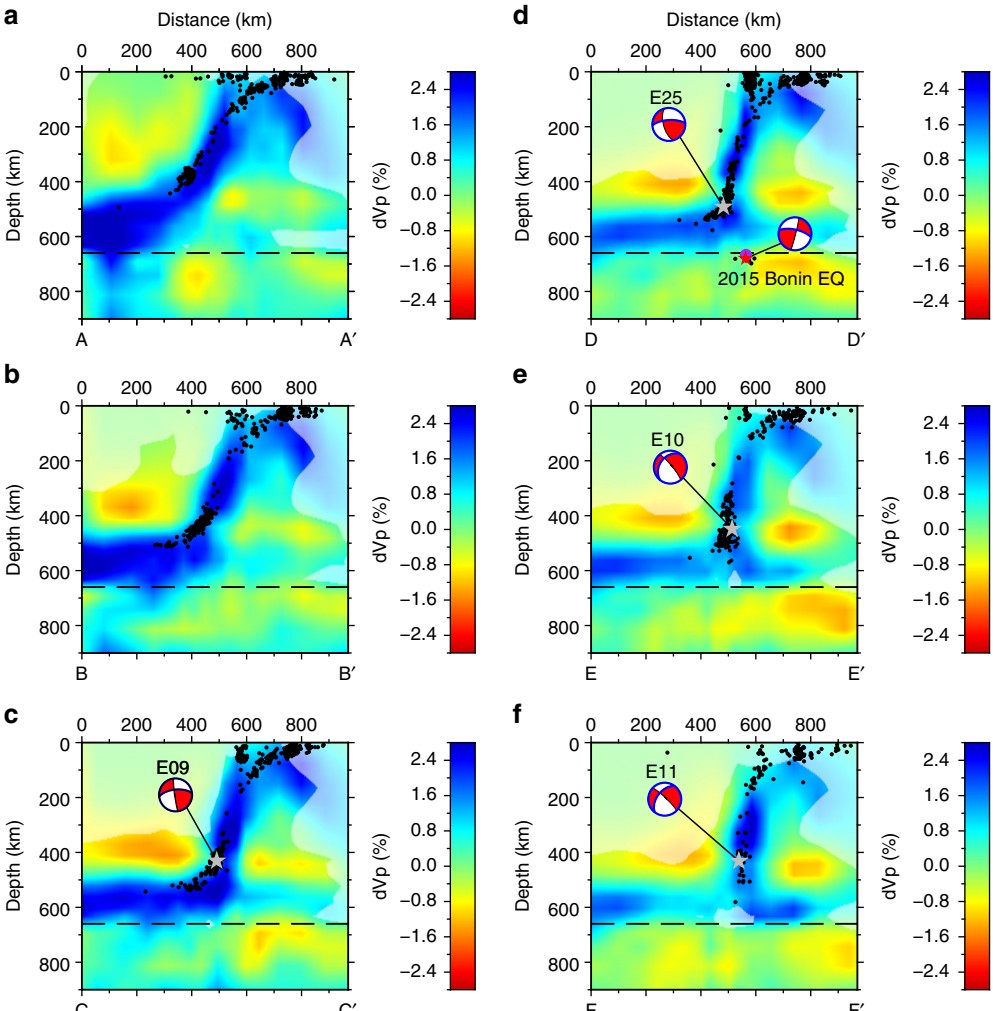

**Fig. 2** Tomographic images of P-wavespeed and relocated earthquakes within the sections shown in Fig. 1. The shaded regions are associated with low model resolution, which is estimated by calculating the semblance values between the true and recovered checkerboard anomalies (see Supplementary Information). Note that in Sections d–f, the high-velocity anomaly appears to split in the mantle transition zone (MTZ). This is mainly due to the orientation of the slab tear. Sections d and e show parts of the slab on both sides of the inferred slab tear (see Figs. 1 and 3). Also shown are focal mechanisms and rupture planes for moderate-to-large earthquakes (Mw > 5.7) in the MTZ determined by directivity analysis in a previous study (Myhill and Warren[43]), rotated into the plane of section. The EXX codes represent the catalogue IDs in that study. The rupture planes for each earthquake (Myhill and Warren[43]; Ye et al.[29]) are highlighted in blue

similar observation has been made in previous tomographic studies[23,40], and interpreted as an approximately 300 km-wide extensional tear in the slab. This interpretation is paradoxical, as this part of the slab is currently in compression[18,41,42]. The higher resolution of our study suggests that the low-wavespeed anomaly is quasi-horizontal, and no more than ~50 km wide. Directivity analysis on several earthquakes[43] contained within the anomaly indicates that co-seismic rupture preferentially takes place on planes aligned with this zone (Fig. 2). This combination of relatively low wavespeeds and evidence of simple shear suggests that the low-wavespeed anomaly may demarcate an incipient shear zone. We propose that the sequence of events, which led to the formation of the shear zone, are as follows: (1) A fold formed in the slab at ~400 km depth (e.g., Myhill[18]). (2) The deep slab met increasing resistance to motion at the bottom of the MTZ. (3) Continued trench advance resulted in the shallower part of the slab advancing over the deeper part of the slab (e.g., Cizkova and Bina[19]; Holt et al.[21]). (4) Localisation of deformation has resulted in formation of a shear zone.

Reductions in P-wavespeed in shear zones could be the result of shear heating (increasing temperatures lowers the wavespeeds of elastic waves) or grain-size reduction (which lowers wave-speeds by increasing grain boundary attenuation[44,45]). Heating could further reduce seismic wavespeeds by creating hydrous fluids through the dehydration of hydrous minerals, which can potentially reside in the cold core of the slab[36]. Other hypotheses for the low-wavespeed anomaly, such as the localised presence of metastable olivine[46], seem less likely, as there is no evidence from plate reconstructions for an increase in lithosphere age or sub-duction velocity in this region, which would increase the pre-servation potential of olivine (see Supplementary Information).

If shear localisation is the cause of the low-wavespeed anomaly, the slab must have a composition and rheology that permits observable reductions in Vp. For example, if the slab were weak, viscous dissipation would not be sufficient to increase the tem-perature of the slab or effectively reduce grain sizes. The tem-perature derivative of P-wavespeed ($\partial V_P/\partial T|_P$) in ultramafic rocks in the deep upper mantle and MTZ is ~−0.4 m/s/K[47], such

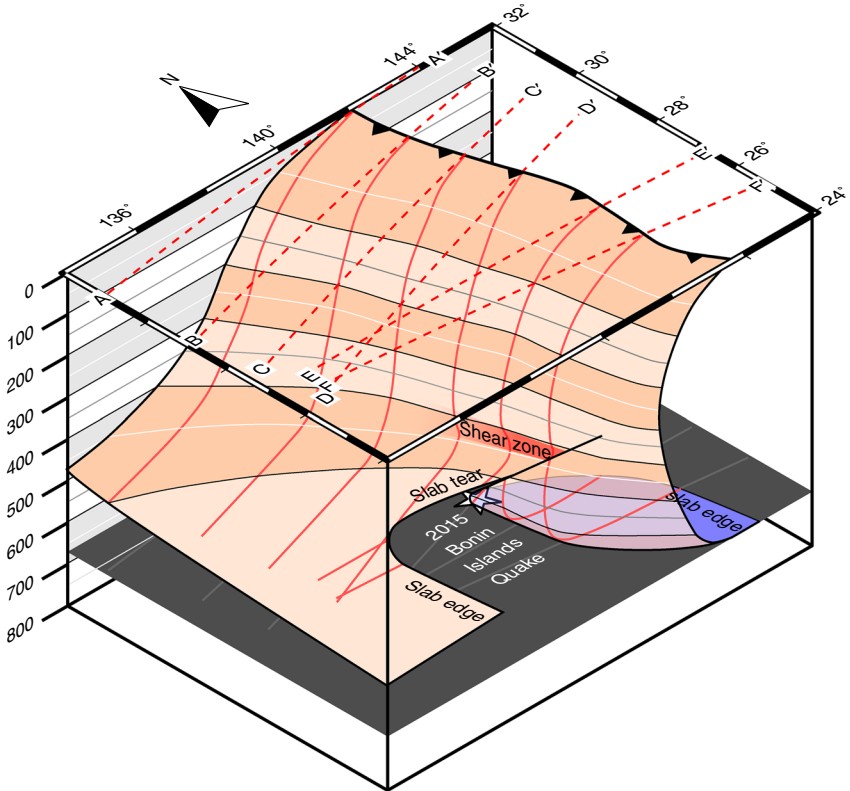

**Fig. 3** Isometric cartoon of the surface of the present-day Izu–Bonin slab based on the tomographic images. The sections shown in Fig. 2 are projected onto the 0 km-depth isocontour (dotted red lines), and onto the surface of the slab (solid red lines). The blue region marks where the slab is overturned. The dark grey surface marks the "660"-km-depth isocontour. The location of the 2015 Bonin Islands earthquake is shown as a white star. Other labelled features correspond to those mentioned in the main text

that a $V_P$ reduction of 1% by shear heating alone corresponds to a temperature increase of ~200 K ($V_P$ ~9 km/s), which must have been generated over at most a few million years. A simple Couette flow model (see Supplementary Information) suggests that this amount of heating requires a viscosity on the order of $10^{23}$–$10^{24}$ Pas. At reasonable strain rates of $10^{-14}$–$10^{-13}$/s, this viscosity is similar to laboratory estimates for dry olivine[48] (see also Supplementary Fig. 22). This high apparent strength is supported by a lack of thickening of the high-wavespeed anomalies in the tomographic cross sections within the upper mantle (Fig. 2).

Our tomographic images place further constraints on the rheology of the slab. Newtonian rheologies (where stress and strain have a simple linear relationship) do not result in the formation of narrow shear zones. Thus, our observations of buckling, tearing and shear zone formation suggest that the Izu–Bonin slab is strongly non-Newtonian. Accurate geodynamic simulations of the region must therefore use material models with nonlinear rheologies[49–51].

The refined shape of the slab also helps to explain a variety of observations of the 2015 Mw 7.9 deep-focus Bonin earthquake. We estimate that the event occurred <30 km south of the tear in the slab, near the fold hinge where the slab rotates from near-vertical to overturned (Figs. 2d and 3). This location explains the lack of a protracted P-wave coda[30] and the lack of ground motion amplification in Japan[52], both of which would have required a continuous high-wavespeed layer extending north of the event to act as a waveguide. The absence of waveform triplications from the "660 km" discontinuity from this event, argued to be because the event was confined to the lower mantle[53], may alternatively be the result of multiple phase transitions associated with strong temperature gradients around the hypocentre (c.f. Cottaar and Deuss[54]). Our interpretation is also largely consistent with the data of Porritt and Yoshioka[26], whose P to S receiver function data indicated that seismic energy passing upwards through the source region of the 2015 earthquake traversed more than one region of high seismic wavespeeds.

There are similarities and also important differences between our interpretation and that of previous studies. We agree with Zhao et al.[25] and Ye et al.[29] (their Model 2) that the slab must be torn, and that the 2015 earthquake ruptured the northern edge of the southern part of the slab. However, in our interpretation, the tear in the slab is not a vertical E–W-striking plane, but instead dips at a moderate angle towards the north. A vertical E–W tear could not explain the presence of the westward-dipping Wadati–Benioff zone about 150 km to the southwest and 150 km above the 2015 Bonin earthquake (Fig. 1). Secondly, we advocate for complete overturn of the slab within the MTZ. Figure 2d–f indicate that the 2015 Bonin Islands earthquake ruptured the lower edge of the imaged high-wavespeed anomaly. Assuming that the highest wavespeeds correspond to the coldest parts of the slab, and using thermal modelling results, which indicate that the coldest parts of the Izu–Bonin slab are about 30 km from the slab surface (e.g., Emmerson and McKenzie[55]), the earthquake occurred <25 km from the crustal section of the slab. This interpretation differs from other analyses of the 2015 Bonin earthquake, which locate the earthquake close to the "bottom" of the subducting slab about 100 km from the crustal section, and therefore in a high-temperature region[25–27,30]. As all potential mechanisms for deep-focus earthquake generation require relatively low temperatures[56], our new interpretation resolves what

was a major problem for theories of deep earthquake generation. Nevertheless, our results indicate that the 2015 Bonin Islands earthquake joins a growing list of large, isolated deep-focus earthquakes, which are believed to lie close to the edges of subducting slabs: Spain 1954; Peru, 1963; Colombia, 1970; Bolivia, 1994; Tonga, 1994[57] and more recently Okhotsk, 2013[58]. For this reason, the 2015 Bonin Islands Earthquake may have ruptured a region experiencing high thermal gradients and more rapid heating than most of the slab.

In this study, we have shown that DD tomography can resolve much finer-scale structures than conventional tomographic techniques. It is particularly suited to imaging tears and other heterogeneities in subducting slabs, which are well illuminated by seismic sources. The new images provide estimates of the relative slab thickening and the kinematics of slab disruption, thus providing much-needed reference data for geodynamic models[51]. In the case of the Izu–Bonin slab, the localised shearing and tearing revealed by our Vp images suggest that the slab is highly viscous, and that nonlinear rheology plays a fundamental role in governing slab deformation and morphology. Finally, the simultaneous relocation of earthquakes during DD inversions provides a robust method to link deep earthquakes directly to the thermal conditions in their source regions.

## Methods

**Teleseismic double-difference tomography.** In this study, we use the teleseismic double-difference tomography software (teletomoDD)[38], which is an implementation of the DD tomography algorithm of Zhang and Thurber[37,61], extended from the regional scale to the global scale. It uses differential arrival times from event pairs observed at global seismic stations to simultaneously determine seismic event locations and wavespeed structure[38]. It can image the source region structure by cancelling the effect of model anomalies away from the source region. It can therefore image the fine-scale structure of a seismically active area even if the coverage of regional stations is sparse.

For each station $k$ that records event pairs $i$ and $j$, the misfits between the observed and predicted arrival times are linearly related to the desired perturbations to the hypocentre location, origin time and wavespeed structure parameters of both global and regional models along the ray path. Mathematically, the relations are as follows:

$$r_k^i = \sum_{l=1}^{3} \frac{\partial T_k^i}{\partial x_l^i} \Delta x_l^i + \Delta \tau^i + \sum_{n\in G} w_n^G \delta u_n^G + \sum_{n\in L}^{ik} w_n^L \delta u_n^L + s_k \quad (1)$$

$$r_k^j = \sum_{l=1}^{3} \frac{\partial T_k^j}{\partial x_l^j} \Delta x_l^j + \Delta \tau^j + \sum_{n\in G} w_n^G \delta u_n^G + \sum_{n\in L}^{jk} w_n^L \delta u_n^L + s_k \quad (2)$$

where $r_k^i$ and $r_k^j$ are the arrival-time residuals for event pair $i$ and $j$ at station $k$, $x$ and $\Delta x$ are hypocentre coordinates and their perturbations, $\Delta \tau^i$ and $\Delta \tau^j$ are the origin time perturbation for event $i$ and $j$, $\delta u_n^G$ and $\delta u_n^L$ are slowness perturbations for global ($G$) and local ($L$) models, respectively. $w_n^G$ and $w_n^L$ are the weighted ray lengths for the global and local model nodes, and $s_k$ is the station correction.

By subtracting Eq. (2) from Eq. (1), we obtain

$$r_k^i - r_k^j = \sum_{l=1}^{3} \frac{\partial T_k^i}{\partial x_l^i} \Delta x_l^i + \Delta \tau^i + \sum_{n\in L}^{ik} w_n^L \delta u_n^L - \left( \sum_{l=1}^{3} \frac{\partial T_k^j}{\partial x_l^j} \Delta x_l^j + \Delta \tau^j + \sum_{n\in L}^{jk} w_n^L \delta u_n^L \right) \quad (3)$$

where $r_k^i - r_k^j = \left(T_k^i - T_k^j\right)^{obs} - \left(T_k^i - T_k^j\right)^{cal}$ is called the double difference. Since teletomoDD includes all possible body-wave arrival and differential time data observed at any distance, a nested regional–global method is utilised to properly constrain wavespeed anomalies along portions of seismic rays outside the regional model[38]. In this method, a finely gridded regional model enclosing the target area is built into a coarser global model (Supplementary Fig. 2). A nonuniform node parameterisation is used for the regional model (Supplementary Fig. 2).

Due to differencing in travel times for pairs of events on common stations, differential arrival times may contain larger random errors. Conversely, the differencing operation can greatly reduce the effect of systematic errors in absolute arrival times[37]. Waveform cross-correlation can be used to determine relative arrival times among events much more accurately than absolute times[62], as long as the two events are close together and have similar focal mechanisms. Finally, in the DD tomographic method, absolute arrival times are used in addition to double-difference measurements, in order to improve absolute event locations[37].

**Modelling viscous dissipation in a slab shear zone.** We consider a simple Couette flow model for a slab shear zone of thickness $w$ (m), where the difference in velocity between the two sides of the zone is equal to $v_x$ (m s$^{-1}$). In such a model, the strain rate is constant within the shear zone:

$$\dot{\varepsilon}_{xz} = \frac{v_x}{w} \quad (4)$$

Neglecting diffusion and phase transitions, and assuming that all energy is dissipated by heat (ignoring processes such as grain-size reduction), the rate of viscous dissipation is

$$\dot{E} = \rho C_p \dot{T} = \sigma_{II} \dot{\varepsilon}_{II} \quad (5)$$

$$\dot{\varepsilon}_{II} = \sqrt{\frac{1}{2}\left(\dot{\varepsilon}_{ii}\dot{\varepsilon}_{jj} - \dot{\varepsilon}_{ij}\dot{\varepsilon}_{ij}\right)} \quad (6)$$

$$\sigma_{II} = \sqrt{\frac{1}{2}\left(\sigma_{ii}\sigma_{jj} - \sigma_{ij}\sigma_{ij}\right)} \quad (7)$$

where $\rho$ and $Cp$ are the density and isobaric heat capacity (approximated as constant, $\rho = 3750$ kg m$^{-3}$ and $Cp = 1200$ J K$^{-1}$ kg$^{-1}$), $\dot{T}$ is the heating rate and $\dot{\varepsilon}_{II}$ and $\sigma_{II}$ are the square roots of the second invariant of the strain rate (s$^{-1}$) and stress (Pa) tensors. The relationship between the stress and strain invariants is given by the following expression:

$$\dot{\varepsilon}_{II} = \sum_{i} A_i \sigma_{II}^{n_i} d^{-p_i} \exp\left(\frac{-(E_{iact} + pV_{iact})}{RT}\right)\left(1 - \left(\frac{\sigma}{\sigma_i}\right)^{q_i}\right)^{s_i} \quad (8)$$

where $p$ is the pressure (Pa), $R$ the gas constant (J K$^{-1}$ mol$^{-1}$), $T$ the temperature (K) and $d$ is the grain size (m). For each flow law $i$ (diffusion, dislocation and Peierls Law creep), $A_i$ is a flow law prefactor, $n_i$ and $p_i$ are constant exponents and $E_{iact}$ and $V_{iact}$ are the activation energy (J mol$^{-1}$) and volume (m$^3$ mol$^{-1}$). Values for these variables are given in the Supplementary Information. Experimental prefactors $A_{expt}$ are often derived from the deviatoric stress ($\sigma_{11} - \sigma_{33}$) and axial strain ($\varepsilon_{11}$) recorded during uniaxial compression experiments. Therefore, a correction is needed to convert these into the prefactors $A_i$:

$$A_i = \frac{3^{\frac{n_i+1}{2}}}{2} A_{iexpt} \quad (9)$$

The derivation of the correction is given in the Supplementary Information (S1.3) of Dannberg et al.[63].

## Code availability

The code teleTomoDD realising teleseismic DD tomography is available upon request.

## Data availability

The EHB catalogue used in this study can be accessed via the website www.isc.ac.uk/isc-ehb.

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

## Acknowledgements

This research is partly supported by the National Natural Science Foundation of China under grant number 41861134009. We are grateful for constructive comments by Rob van der Hilst, Rob Porritt, Zhongwen Zhan and an anonymous reviewer.

## Author contributions

H.Z. designed and supervised the project. H.Z. and R.M. wrote the paper and made interpretations. F.W. assembled data, conducted tomographic inversions and synthetic tests. H.G. contributed to assembling data and conducting tomographic inversions.

## Additional information

**Competing interests:** The authors declare no competing interests.

