## [Peer review file · Nature Communications]

Reviewers' comments:

Reviewer #1 (Remarks to the Author):

The paper shows a P-wave velocity image of the Izu-Bonin subduction system obtained from double difference teleseismic tomography. The authors then use this velocity model and additional forward modelling to infer the detailed slab morphology and to put constraints on different deformation mechanisms that may be active. As their dataset also contains a 2015, M7+ deep-focus earthquake, the double-difference tomographic approach allows to precisely relocate its hypocenter relative to the high velocity slabs.

I think this is an interesting paper which has the potential to add significant advances in the field of earth sciences. However, I have several concerns regarding its present structure:

Major concerns:

- As the tomographic velocity model forms the base of this paper, its resolution limits and other limitations should be discussed in more detail. Specifically, end member synthetic tests are missing, which would clearly show how the inferred slab tear can be resolved. Further, I was surprised that the one synthetic test shown creates many artefacts with negative polarity. What is the origin of these artefacts? If this is not explained in a satisfying way, the synthetic tests rather decrease than increase my confidence in the real data.
- In the interpretation I had partly problems to follow the line of argumentation: many arguments are based on 'observations' on the velocity model that did not convince me (e.g. how robust is the 1% velocity reduction that is used as a base for further calculations?).
- As there exists already many other publications on the Bonin-Island earthquake and the velocity structure of the slab, I had the feeling that this work should be acknowledged in more detail. If the authors state that their conclusion is different, they should discuss more carefully why they think so (e.g. the overturning of the slab is already suggested in Zhao et al. 2017. Thus the authors have to show clearer how robust this feature is in their inversion and why they think their images are better. They could show how the event location is shifted during the inversion; they could put the earthquake at an artificial location and show how it is relocated; they could show on what data the location depends on; etc. ...).

Please find more details on these and other points below:

Line 29: what is a 'plunging tear'? Possibly add attributes like "horizontally" or "vertically"

Line 33: I don't like the word "recumbent" as the velocity image is static not dynamic

Line 67: please be more specific instead of writing 'fine-scaled'

Line 71: for anybody not familiar with this subduction system: it would be great to describe at least once where it is located. E.g. 'XX km NE of Japan'

Line 73-75: Please be more specific: in which way was the morphology influenced?

Line 77: here, it would be more informative to explain in which ways the tomographically imaged velocity anomalies differ in these different studies. Are the interpretations different because the velocity models are very different or are the models very similar but there is a disagreement on how to interpret them?

Line 116: 200 km vs. 30-40 km. This is a very asymmetric resolution. Possibly add a ray coverage plot (could be added in S5 or S6). Is the ray distribution that asymmetric? Further, some of the cross sections in Figure 2 have less than 200 km distance – Does it make sense to plot cross-sections denser than the resolution limit?

Line 116: Where is the information on the resolution limit taken from? I cannot find these values e.g. in the supporting information. Is it 'fit by eye' by looking at the synthetic tests?

Line 117: I am not convinced by the resolution of the model. Possibly add here the conclusions drawn from the supporting information: '[...] the geometry is resolved, but amplitudes are underestimated. Additionally negative anomalies may be created artificially due to XX [...] – why are these negative anomalies created?

Line 152: Isacks & Molnar (1971) is a global compilation of different subduction zones. The

"compression" referred here to was derived based on two earthquakes that occurred between 1962 and 1968. I am wondering if there exists stronger and newer evidence. --- Especially as the earthquakes in your dataset do not indicate compression exclusively.

Line 158: please don't just put this buzzwords. Please shortly explain why you think each of them would reduce the velocity.

Line 159: 'the location ..'  'the location of the anomaly ..'

Line 159: Do you really mean the bottom of the mantle or do you mean the transition zone?

Line 160: the Holt et al 2017 citation is missing in the reference list

Line 160: 'In this interpretation ..'  In their interpretation or in your one? On what is this interpretation based? Data? Modelling? ..

Line 160-162: this sentence seems grammatically wrong

Line 162-164: in this sentence the grammar seems very complex to me. Possibly split into two. What is "it" referring to?

Line 173-176: this sentence is hard to understand. Possibly split into two. Please add a reference or explanation. Why is the transformation enhanced by shear?

Lines 178-206: I suggest to reformulate or to join these two paragraphs. It was hard to follow the line of argumentation.

Line 179: why must the rheology permit a reduction in v_p in this case? You mean because both effects are too much?

Line 189: Does this sentence now refer to the slab-gap? The relation is not quite clear as you write about a "lack in thickening" in the sentence before.

Lines 190-192: Add references. Such an effect of grain size reduction was shown by Thielmann et al., 2015, for instance. What would be the fluid source you are suggesting here?

Lines 190-192: This is the first time you talk about a non-Newtonian rheology. The line of argumentation is not clear to me.

Line 195: please make clear that these authors investigate the buildup of two subduction systems with similar subduction polarity and some distance in between. "Double subduction" could also be slab-slab interaction (which was investigated a lot in the Faccenna group as well).

Line 196: I don't understand why the geodynamic models are used as 'prove' for the tomographic model shown here. The geodynamic models have been designed based on tomographic images - isn't this a chicken-and-egg argumentation?

Line 196: this is the first time the Ryukyu subduction system is mentioned. It is not mentioned in the introduction or in your figures. Please make sure that also a reader not familiar with the tectonics of the study region understands this paragraph.

Line 202: I don't understand this argumentation. You mean a tear in the slab is necessary to explain the different amount of deformation in the north and south?

Line 204: Possibly indicate in Figure 1 which feature you are referring to. You mean that the entire slab broke-off deeper than the anomaly imaged? Would the implication be a longer subduction history than thought before? Or did I possibly misunderstand this sentence?

Line 217: Zhao et al. (2017) relocated the earthquake also in a newly obtained regional velocity model. Although it is not a joint inversion, their results should be quite reliable as well. Here would be a good point to discuss on these different locations and interpretations.

Line 241: I disagree. Based on the data and synthetic tests shown I am not convinced that e.g.

tears can be well resolved. I suggest to show end member tests to prove this statement.

Line 242: better: 'relative slab thickening' as you write above that the absolute thickness cannot be estimated.

Line 243: 'tests' is a strange word when speaking about real data. What about "reference dataset" or something similar?

Line 244: "vp images show"  better: 'suggests' as it is not an observation but your interpretation

Line 245: For me it is still not clear: what exactly are the authors' arguments for non-linear rheology?

Line 248: why do you not write here at which location you obtain the earthquake hypocenter? Isn't this one of the main findings?

Line 248-251: I am not sure how these two last sentences should add any information to the conclusion (except advertising another paper by the first author)? -- Sorry -- I suggest to remove these sentences

Figures:

Figure 1: Possibly scale the earthquakes by magnitude and specify in the caption which time of EHB data is plotted.

Figure 2: Why is figure a-c cut to the west? It would be nice to see how the slab continues to the west (until the resolution limit is reached)

What do you mean with "representative earthquakes"? It would be more useful to plot e.g. the largest earthquakes, possibly also highlighting their aftershock sequences.

It is unclear to me how the resolution limit is determined

In the text you refer to the sub-figures with capital letters; in the figure they have small letters - please unify this.

Figure 3: 'the shear zone'  'the inferred shear zone'

Why is the overturned part blue? Should this be a shadow?

Please write at some point that the green area is the 660.

Supporting information:

Line 41: How are the outliers removed? Visually? Based on some mathematical criterion?

Line 43: Is the maximum epicentral distance 100 degree? Is there any special treatment for Pdiff?

Line 47: an average offset of 54 km seems quite a lot to me. What are the maximum/minimum values? Did you put any constraints on these? Did you do tests with smaller/larger offsets?

Line 169: '180km'  '180 km'

Lines 94-95: Isn't this conclusion on the resolution limit a little bit over-simplified? The resolution also varies laterally. It is still not clear to me based on what information the very complex resolution limit in Figure 2 is constrained.

Line 105: I am quite surprised to see so much negative energy in the recovered velocity model. Where does it come from? Is the corresponding energy mapped in station or event terms?

Line 179: Full stop missing.

Figure S1: Please add ellipses to indicate epicentral distance from study region.

Figure S4: This is an irritating representation for the checkerboard tests – please add an explanation in Figure caption: are the rectangles the values at each node? Possibly show in the same interpolation than the real data? Use same color scheme and model geometry representation than for S5 and S6.

Figure S6: Please add the geometry of the input anomaly on top – otherwise it is very hard to see differences. Possibly add the hypocenters prior to inversion to see their shift. Use exactly the same color scheme than in S5.

Figure S5: I think it is irritating to choose such a complex input synthetic model. It is hard to see which features are recovered, smeared etc.. I suggest to design an easier model (e.g. cubic forms with constant amplitudes) which has the characteristics of the real data.

Figures S5/S6: Where are Figs. S5 and S6 located on the map? The same nomenclature as in Fig. S4 should be used.

Figures S5/S6: In the main text you write that you estimate the resolution to be ~ 200 km in N-S? Where does this information come from? Please plot horizontal sections here as well - it's hard to understand otherwise.

Figure S6 exists twice!

Figure S7: Label a,b,c are missing. The annotation of the y axes is missing.

Reviewer #2 (Remarks to the Author):

In this manuscript, the authors present tomographic images of the subducting slab in the Izu-Bonin subduction zone and evaluate the relationship of the imaged structure to the 30 May 2015 deep earthquake. This earthquake is extremely unusual due to its depth, magnitude, and distance from the Wadati-Benioff zone and therefore understanding the processes leading to this earthquake provide rare insights into deformation near the base of the mantle transition zone.

Their model is significantly higher resolution than previous models due to their double difference methodology, which has maximum resolution in regions of high seismic activity. I would appreciate a discussion (briefly in the main text and slightly less brief in the supplements) concerning the weaknesses in the method. For instance, I would assume that due to the double differencing, the results can be more skewed than traditional body wave tomography by outliers. This is why careful rejection of outliers (e.g. Fig. S2) is so important. I would also like to know why the spatial resolution in latitude is an order of magnitude worse than in depth and longitude. This seems odd and comes back in line 153 when they discuss that the low velocity anomaly is no more than ~ 50 km wide and it is unclear in which direction this refers. This also leads me to ask how the mesh inversion mesh was defined. Did the authors simply say that this is the region of interest and has high seismicity or did they use a data adaptive meshing algorithm (this is probably a supplementary information topic)?

The primary findings of the model indicate 4 major regions. The northernmost region is a steeply dipping slab, which flattens in the transition zone. South of that, the slab has the same geometry, but a horizontal tear is developing which is separating the steeply dipping slab from the slab lying flat in the transition zone. Further south, between 26.5° and 28.5° , is a large-scale plunging tear. The southernmost section, south of the tear is where the slab is overturned in the lower mantle with the top of the slab lying on the base of the transition zone. It is in this, southernmost overturned part of the slab, that the earthquake occurred. Due to the apparently relatively low resolution in latitude, it is unclear how well separated these regions are. A resolution map or north-south cross section may clarify this uncertainty.

Considering the scatter in ISC data, I'm curious how much the structure is dependent upon that

data. There are two types of tests that could be utilized for this. First, rerun the inversions using other publicly available datasets. The first dataset that I can think of is the Array Network Facility, which provides USArray delay times. That will likely have bias due to the limited aperture of the array but could be combined with other datasets. An alternative would be to bootstrap resample the travel times and see how much that causes the structure to vary.

There is something unclear in regards to the geometry of the slab as illustrated in Figure 3 and Figure 2e,f. The horizontal slab in the transition zone could be inferred to lying both east and west. This is similar to the uncertainty in geometry shown by Obayashi et al., 2013 and is dependent upon your resolution in latitude.

Line 181: add parentheses around -0.4 to separate it from the \sim symbol.

The supplementary information could use careful proofreading. It is mostly good, but there are a few confusing sentences.

I would also ask the authors, if possible, that they make the 3D Vp model publicly available as research into this earthquake will likely be ongoing.

Overall, I find this to be a good paper, requiring only minor revisions. I don't see this as inherently contradictory to my earlier work on the region, but rather I think this higher resolution model is consistent with the data I presented and sheds valuable light onto the ongoing processes of the region.

Rob Porritt

References:

Obayashi, M., J. Yoshimitsu, G. Nolet, Y. Fukao, H. Shiobara, H. Sugioka, H. Miyamachi, and Y. Gao (2013), Finite frequency whole mantle P wave tomography: Improvement of subducted slab images, *Geophys. Res. Lett.*, 40, 5652–5657, doi:10.1002/2013GL057401.

We would like to sincerely thank Dr. Porritt and one anonymous reviewer for the constructive comments. We marked the original comments in **bold**, our replies in regular, and newly added text in the revised paper in *italic* fonts, respectively.

Reviewers' comments:

Reviewer #1 (Remarks to the Author):

The paper shows a P-wave velocity image of the Izu-Bonin subduction system obtained from double difference teleseismic tomography. The authors then use this velocity model and additional forward modelling to infer the detailed slab morphology and to put constraints on different deformation mechanisms that may be active. As their dataset also contains a 2015, M7+ deep-focus earthquake, the double-difference tomographic approach allows to precisely relocate its hypocenter relative to the high velocity slabs.

I think this is an interesting paper which has the potential to add significant advances in the field of earth sciences. However, I have several concerns regarding its present structure:

Reply: We greatly appreciate your detailed and insightful comments that are very helpful for improving our paper. Thanks you! This really shows the added values of peer review.

Major concerns:

- As the tomographic velocity model forms the base of this paper, its resolution limits and other limitations should be discussed in more detail. Specifically, end member synthetic tests are missing, which would clearly show how the inferred slab tear can be resolved. Further, I was surprised that the one synthetic tests shown creates many artefacts with negative polarity. What is the origin of these artefacts? If this is not explained in a satisfying way, the synthetic tests rather decrease than increase my confidence in the real data.

Reply: Thanks for your suggestions. We agree that more discussions and tests of tomographic models are very important to show the reliability of our inverted models. For this purpose, we have tested a series of models including end member slab models. In addition to checkerboard tests, we have conducted a restoration resolution test using the inverted model to create synthetic times. We also conducted bootstrapping analysis to show the uncertainties for the inverted model are small. All of these tests show that the main slab features that we discussed can be well-resolved by our assembled data and the DD tomographic method.

For the artefacts shown in the original synthetic test results, they are actually caused by the mistake in setting velocity values on grid nodes below 660 km and the inappropriate setting of

colorbar. In the revised supplementary information, we have corrected this mistake and use a more appropriate colorbar. Now inverted models are very close to true synthetic models.

- In the interpretation I had partly problems to follow the line of argumentation: many arguments are based on 'observations' on the velocity model that did not convince me (e.g. how robust is the 1% velocity reduction that is used as a base for further calculations?).

Reply: With much more tests added in the supplementary information, the main slab features that were discussed are shown to be robust.

- As there exists already many other publication on the Bonin-Island earthquake and the velocity structure of the slab, I had the feeling that this work should be acknowledged in more detail.

Reply: We have now acknowledged these studies (and particularly Zhao et al., 2017) in more detail.

If the authors state that their conclusion is different, they should discuss more carefully why they think so (e.g. the overturning of the slab is already suggested in Zhao et al. 2017. Thus the authors have to show clearer how robust this feature is in their inversion and why they think their images are better. They could show how the event location is shifted during the inversion; they could put the earthquake at an artificial location and show how it is relocated; they could show on what data the location depends on; etc. ..).

Reply: In the revised supplementary information, we have done different tests to show main features such as slab overturn real. We also more clearly state the differences between our tomographic results and Zhao et al. (2017) and other results, as follows:

"We agree with Zhao et al. (2017) and Ye et al. (2016; Model 2) that the slab must be torn, and that the 2015 earthquake ruptured the northern edge of the southern part of the slab. However, in our interpretation the tear in the slab is not a vertical E-W-striking plane, but instead dips at a moderate angle toward the north. A vertical E-W tear could not explain the presence of the westward-dipping Wadati-Benioff zone about 150 km to the southwest and 150 km above the 2015 Bonin earthquake (Figure 1). Secondly, we advocate for complete overturn of the slab within the mantle transition zone. Figures 2d-2f indicate that the 2015 Bonin Islands earthquake ruptured the lower edge of the imaged high velocity anomaly. Assuming the highest velocities correspond to the coldest parts of the slab, and using thermal modelling results which indicate the coldest parts of the Izu-Bonin slab are about 30 km from the slab surface (e.g. Emmerson and McKenzie, 2007), the earthquake occurred <25 km from the crustal section of the slab. This interpretation differs from other analyses of the 2015 Bonin earthquake, which locate the earthquake close to the "bottom" of the subducting slab about 100 km from the crustal section, and therefore in a high temperature region (Porritt and Yoshioka, 2016; Takemura et al., 2016; Obayashi et al., 2017; Zhao et al., 2017). As all potential mechanisms for deep-focus earthquake generation require relatively low

temperatures (Frohlich, 2006), our new interpretation resolves what was a major problem for theories of deep earthquake generation. Nevertheless, our results indicate that the 2015 Bonin Islands earthquake joins a growing list of large, isolated deep-focus earthquakes which are believed to lie close to the edges of subducting slabs: Spain 1954; Peru, 1963; Colombia, 1970; Bolivia, 1994; Tonga, 1994 (Green and Houston, 1995); and more recently Okhotsk, 2013 (Zhan et al., 2014). For this reason, the 2015 Bonin Islands Earthquake may have ruptured a region experiencing high thermal gradients and more rapid heating than most of the slab..”

Please find more details on these and other points below:

Line 29: what is a ‘plunging tear’? Possibly add attributes like “horizontally” or “vertically”

Reply: The sentence has been rephrased. We now refer to the tear as dipping moderately to the north.

Line 33: I don’t like the word “recumbent” as the velocity image is static not dynamic

Reply: The word recumbent has been changed to flat.

Line 67: please be more specific instead of writing ‘fine-scaled’

Reply: This sentence has been removed, with quantitative descriptions provided at the end of the next paragraph.

Line 71: for anybody not familiar with this subduction system: it would be great to describe at least once where it is located. E.g. ‘XX km NE of Japan’

Reply: We have now added more detail of the geographic location of this subduction zone, as follows:

“The Izu-Bonin subduction zone (Figure 1) is a thousand kilometer long system which extends south from Tokyo, where the Pacific Plate subducts beneath the Philippine Sea Plate. At the northern end of the zone, the Izu-Bonin Trench meets the Japan Trench and the Sagami Trench at the Boso Triple Junction. At the southern end (east of the Volcano Islands), the strike of the Izu-Bonin trench rotates from north-south to northwest-southeast, marking the start of the Marianas Trench.”

Line 73-75: Please be more specific: in which way was the morphology influenced?

We have now elaborated on this sentence to describe the effect of the Philippines slab/trench motions on slab shape. The newly added text is as follows:

“After subduction at the trench, the Izu-Bonin slab descends westward into the upper mantle and transition zone. At 100–400 km depth, the distribution of earthquakes indicates that the dip of the

slab increases from 40° in the north to 80° in the south (Burbach and Frohlich, 1986). This increase in slab dip has been attributed to a higher velocity of trench advance in the south relative to the north over the last few million years (Carlson and Melia, 1984; Le Pichon and Huchon, 1987; Funiello et al, 2008). At greater depths, earthquake locations indicate that the dip of the slab decreases to 20–30° along a fold hinge that deepens toward the south from ~400 km depth at 33°N to ~550 km depth at 27°N (Myhill, 2013). Geodynamic modelling suggests that this morphology may be attributable to westward subduction of the Philippine Sea Plate at the Ryukyu Trench, west of the Izu-Bonin slab (Čížková and Bina, 2015; Faccenna et al., 2017; Holt et al., 2017). Subduction at the Ryukyu trench induces a slab pull on the shallow part of the Izu-Bonin slab, and a positive dynamic pressure between the two subduction systems in the upper mantle and transition zone. These two influences should encourage the Izu-Bonin slab to steepen and become increasingly convex in the direction of subduction in the upper mantle. In the mantle transition zone, some simulations indicate buckling of the slab, while others indicate overturn (Čížková and Bina, 2015; Holt et al., 2017), such that the tip of the slab can either face in, or opposite to, the direction of subduction at the trench.”

Line 77: here, it would be more informative to explain in which ways the tomographically imaged velocity anomalies differ in these different studies. Are the interpretations different because the velocity models are very different or are the models very similar but there is a disagreement on how to interpret them?

Reply: This is because tomographic models are different in different studies, leading to different interpretations. We have made this point clear in the revised text, as follows:

“High resolution seismic tomography should be able to answer these questions, but existing studies have produced a variety of differing tomographic models and thus different interpretations of the deep slab. For example, tomographic images of the southern end of the Izu-Bonin slab have variously been interpreted as evidence for direct penetration into the lower mantle (van der Hilst and Seno, 1993; Miller et al, 2005, 2006; Zhao et al, 2017), multiple isoclinal folds (Porritt and Yoshioka, 2016) or “heel”-shape thickening (Obayashi et al, 2017).”

Line 116: 200 km vs. 30-40 km. This is a very asymmetric resolution. Possibly add a ray coverage plot (could be added in S5 or S6). Is the ray distribution that asymmetric? Further, some of the cross sections in Figure 2 have less than 200 km distance – Does it make sense to plot cross-sections denser than the resolution limit?

Reply: To address this concern, we have re-conducted our tomographic inversion by a two-step approach by first using a coarser inversion grid and then a finer inversion grid. The grid nodes in longitude and depth are the same as before but are finer in latitude with a grid interval of 1° to 2°. Around the 2015 Bonin earthquake, the resolution is 1°. The main features stay the same but the anomaly amplitudes increase. The main purpose of this study is to show the slab morphology from north to south. For this reason, we set up denser grid nodes in the longitude direction, normal to the

trench. For cross sections shown in Figure 2, they are coarser than the actual inversion grid in latitude. These profiles are not along latitude, but rather normal to the local trench strike. For this reason, the velocity profile along each cross section is interpolated from the inverted model.

Line 116: Where is the information on the resolution limit taken from? I cannot find these values e.g. in the supporting information. Is it 'fit by eye' by looking at the synthetic tests?

Reply: We have tried different grid intervals in latitude and longitude by trial and error. The general strategy for selecting inversion grid is to start the coarser grid and gradually try the finer grid. Checkerboard resolution tests can be used to check different inversion grids to see whether the model resolutions are satisfactory. In the supplementary information, we show checkerboard test results for three different grid intervals in latitude (with grid the unchanged in longitude and depth). From these checkerboard tests, it is found that the current grid choice gives satisfactory recovery in checkerboard models, and offers a good compromise between resolution and robust retrieval of velocity anomalies. We have made this point clearer in the revised supplementary information.

Line 117: I am not convinced by the resolution of the model. Possibly add here the conclusions drawn from the supporting information: '[..] the geometry is resolved, but amplitudes are underestimated. Additionally negative anomalies may be created artificially due to XX [...]' - why are these negative anomalies created?

Reply: Thanks for your comment. To address your concern on model resolution, we have conducted more synthetic tests (see the revised supplementary information) and we are confident that the slab morphology shown in this study is real and robust. We agree with your comment that the amplitudes are very likely underestimated due to various regularizations applied to the tomographic system, such as smoothing and damping. The additional apparent negative anomalies shown in Figure S4 of the original supplementary materials were primarily the result of using an inappropriate color scale [-2.5%, 5%]. The plot showing the synthetic model had a wider scale [-5%, 5%]. Because of this, negative anomalies were enhanced. Another reason is that the velocity model at one grid node below the slab was given wrong value when doing the synthetic test. This mistake has been corrected in the new synthetic tests. It is true that the recovered velocities appear slightly slower above and below the slab, but these anomalies are insignificant.

We have added in the main text the following sentence:

“Resolution tests show that the slab is robustly resolved by the data but velocity amplitudes may be underestimated due to smoothing and damping regularizations applied to the DD the tomographic system (see Supplementary Information).”

Line 152: Isacks & Molnar (1971) is a global compilation of different subduction zones. The "compression" referred here to was derived based on two earthquakes that occurred between 1962 and 1968. I am wondering if there exists stronger and newer evidence. ---

Especially as the earthquakes in your dataset do not indicate compression exclusively.

Reply: We've added citations to Alpert et al. (2010) and Myhill (2013), which both show in plane compression as the dominant mechanism.

Line 158: please don't just put this buzzwords. Please shortly explain why you think each of them would reduce the velocity.

Reply: We have now added text explaining (with citations) why heating, grain size reduction and fluid liberation can cause velocity reductions. The revised text is as follows:

“P-wave velocities could be reduced by heating (which lowers the velocities of purely elastic waves) or grain size reduction (which lowers velocities by increasing grain boundary attenuation; Faul and Jackson, 2005, Takei et al., 2014). Heating could further reduce seismic velocities by creating hydrous melts through the dehydration of hydrous minerals, which can potentially reside in the cold core of the slab throughout the upper mantle and transition zone (Omori et al., 2004). Other hypotheses for the low velocity anomaly such as the localised presence of metastable olivine (Jiang et al., 2008) seem less likely, as there is no evidence for an increase in lithosphere age or subduction velocity which would increase the preservation potential of olivine in this region (e.g. Emmerson and McKenzie, 2007, also see Supplementary Information).”

Line 159: ‘the location ..’  ‘the location of the anomaly ..’

Reply: This sentence has now been completely rewritten.

Line 159: Do you really mean the bottom of the mantle or do you mean the transition zone?

Reply: Thank you! We have now corrected this sentence as follows:

“The location is consistent with resistance to motion of the slab at the bottom of the mantle transition zone during trench advance (e.g. Holt et al., 2017).”

Line 160: the Holt et al 2017 citation is missing in the reference list

Reply: This reference has now been added.

Line 160: ‘In this interpretation ..’  In their interpretation or in your one? On what is this interpretation based? Data? Modelling? ..

Reply: This is our interpretation, based on the available geophysical data and geodynamic modelling efforts. We've rephrased the sentence to make this clear.

Line 160-162: this sentence seems grammatically wrong

Reply: This sentence has been replaced.

Line 162-164: in this sentence the grammar seems very complex to me. Possibly split into two. What is “it” referring to?

Reply: The explanation has been replaced by a numbered list to make the sequence of events clearer:

“We propose that the sequence of events which led to the formation of the shear zone are as follows: 1) A fold formed in the slab at ~400 km depth (e.g. Myhill, 2013). 2) The deep slab met increasing resistance to motion at the bottom of the mantle transition zone. 3) Continued trench advance resulted in the shallower part of the slab advancing over the deeper part of the slab (e.g. Cizkova and Bina, 2015; Holt et al., 2017). 4) Localization of deformation has resulted in formation of a shear zone.”

Line 173-176: this sentence is hard to understand. Possibly split into two. Please add a reference or explanation. Why is the transformation enhanced by shear?

Reply: This is actually superfluous to the point we’re trying to make, so we’ve removed the sentence. Appropriate references would have been Dupas-Bruzek et al. (1998) and Mosenfelder et al. (1998, 2000, 2001).

Lines 178-206: I suggest to reformulate or to join these two paragraphs. It was hard to follow the line of argumentation.

Reply: We have moved the rejected hypotheses to Supplementary information, as we couldn’t see a way to improve the flow while keeping all the information. The second paragraph has been reworked into other parts of the text. The new text now reads:

“If shear localisation is the cause of the low-velocity anomaly, the slab must have a composition and rheology that permits observable reductions in V_p . For example, if the slab were weak, viscous dissipation would not be sufficient to raise the temperature of the slab or effectively reduce grain sizes. The temperature derivative of P-wave velocities ($\partial V_p / \partial T|_p$) in ultramafic rocks in the deep upper mantle and mantle transition zone is approximately -0.4 m/s/K (Stixrude and Lithgow-Bertelloni, 2011), such that a V_p reduction of 1% by shear heating alone corresponds to a temperature increase of ~200 K ($V_p \sim 9$ km/s), which must have been generated over at most a few million years. A simple Couette flow model (see Supplementary Materials) suggests that this amount of heating requires a viscosity on the order of 10^{23} - 10^{24} Pas. At reasonable strain rates of 10^{-14} - 10^{-13} /s, this viscosity is similar to laboratory estimates for dry olivine (Supplementary Information; Figure S20). This high apparent strength is supported by a lack of thickening of the high velocity anomalies in the tomographic cross sections within the upper mantle (Figure 2).

Our tomographic images place further constraints on the rheology of the slab. Newtonian rheologies (where stress and strain have a simple linear relationship) do not result in the formation of narrow shear zones. Thus, our observations of buckling, tearing and shear zone formation suggest that the Izu-Bonin slab is strongly non-Newtonian. Accurate geodynamic simulations of the region must therefore use material models with nonlinear rheologies (Billen and Hirth, 2007; Garel et al., 2014; Yang et al. 2017)."

Line 179: why must the rheology permit a reduction in v_p in this case? You mean because both effects are too much?

Reply: We've added an example explaining that for viscous dissipation to reduce seismic velocities, the slab must have a relatively high strength. See the quote above.

Line 189: Does this sentence now refer to the slab-gap? The relation is not quite clear as you write about a "lack in thickening" in the sentence before.

Reply: This sentence was unclear, and has been removed. The previous sentence now includes the words "within the upper mantle". Again, see the quote above.

Lines 190-192: Add references. Such an effect of grain size reduction was shown by Thielmann et al., 2015, for instance. What would be the fluid source you are suggesting here?

Reply: The fluid source would be dehydration of hydrous minerals. A reference to Omori et al. (2004) is now included earlier in the text (see above).

Lines 190-192: This is the first time you talk about a non-Newtonian rheology. The line of argumentation is not clear to me.

Reply: We have rewritten this part of the text. See the second paragraph, quoted above.

Line 195: please make clear that these authors investigate the buildup of two subduction systems with similar subduction polarity and some distance in between. "Double subduction" could also be slab-slab interaction (which was investigated a lot in the Faccenna group as well).

Reply: This line has now been clarified, as follows:

"The slab morphology observed in our tomographic images closely resembles the morphologies produced by geodynamic simulations which model two closely-spaced (<3000 km) slabs subducting with the same polarity (Čížková and Bina, 2015; Faccenna et al., 2017; Holt et al., 2017), like the

Izu-Bonin and Philippines slabs. These simulations impose similar plate velocities to those seen in the Izu-Bonin-Philippines region, but do not use output from tomographic inversions as input to the simulations.”

Line 196: I don't understand why the geodynamic models are used as 'prove' for the tomographic model shown here. The geodynamic models have been designed based on tomographic images - isn't this a chicken-and-egg argumentation?

Reply: The geodynamic models – particularly those of Holt et al. (2017) – are not designed based on tomographic images, but instead use boundary conditions (plate velocities) similar to those obtained from GPS / plate reconstructions. We have rephrased the paragraph to make this more clear, as follows:

*“The slab morphology observed in our tomographic images closely resembles the morphologies produced by geodynamic simulations which model two closely-spaced (<3000 km) slabs subducting with the same polarity (Čížková and Bina, 2015; Faccenna et al., 2017; Holt et al., 2017), like the Izu-Bonin and Philippines slabs. These simulations impose similar plate velocities to those seen in the Izu-Bonin-Philippines region, **but do not use output from tomographic inversions as input to the simulations.**”*

Line 196: this is the first time the Ryukyu subduction system is mentioned. It is not mentioned in the introduction or in your figures. Please make sure that also a reader not familiar with the tectonics of the study region understands this paragraph.

Reply: Ryukyu has now been mentioned in the introduction:

“Geodynamic modelling suggests that this morphology may be attributable to westward subduction of the Philippine Sea Plate at the Ryukyu Trench, 2000–3000 km west of the Izu-Bonin slab (Čížková and Bina, 2015; Faccenna et al., 2017; Holt et al., 2017). Subduction at the Ryukyu trench induces a slab pull on the shallow part of the Izu-Bonin slab, and a positive dynamic pressure between the two subduction systems in the upper mantle and transition zone. These two influences should encourage the Izu-Bonin slab to steepen and become increasingly convex in the direction of subduction in the upper mantle.”

Line 202: I don't understand this argumentation. You mean a tear in the slab is necessary to explain the different amount of deformation in the north and south?

Reply: Yes. However, this is difficult to demonstrate without referring to the figure in Cizkova and Bina (2015), so we have elected to remove the sentence. The beginning of the discussion should now explain the argument more clearly:

“The abrupt change in orientation of the high velocity anomalies in the mantle transition zone seen in Sections CC'–EE' requires the presence of a narrow slab tear at least 500 km long. Our interpretation

is shown in Figure 3. The northern edge of the tear marks the southern termination of the sub-horizontal slab, while the southern edge of the tear dips at an angle of ~45 degrees to the NNE, beneath the bend in the main Wadati-Benioff zone. The tear decouples the northern and southern parts of the deep slab, allowing the southern end to overturn, making Izu-Bonin the first reported subduction zone where the slab lies inverted on top of the upper-lower mantle boundary.”

Line 204: Possibly indicate in Figure 1 which feature you are referring to. You mean that the entire slab broke-off deeper than the anomaly imaged? Would the implication be a longer subduction history than thought before? Or did I possibly misunderstand this sentence?

Reply: We have extended the western limit of our inversion. Figure 3 now shows the proposed tear in the slab. The text now reads as follows:

“The overturned slab fragment in Figure 2f appears to be about 300 km long. We propose that the relatively low velocities observed over a similar distance further west in the same cross section correspond to the slab gap created during tearing (Figure 3). A similar interpretation was made by Zhao et al. (2017).”

Line 217: Zhao et al. (2017) relocated the earthquake also in a newly obtained regional velocity model. Although it is not a joint inversion, their results should be quite reliable as well. Here would be a good point to discuss on these different locations and interpretations.

We have added a paragraph describing the similarities and differences between our interpretation and that of other studies, including Zhao et al. (2017):

“There are similarities and also important differences between our interpretation and that of previous studies. We agree with Zhao et al. (2017) and Ye et al. (2016; Model 2) that the slab must be torn, and that the 2015 earthquake ruptured the northern edge of the southern part of the slab. However, in our interpretation the tear in the slab is not a vertical E-W-striking plane, but instead dips at a moderate angle toward the north. A vertical E-W tear could not explain the presence of the westward-dipping Wadati-Benioff zone about 150 km to the southwest and 150 km above the 2015 Bonin earthquake (Figure 1). Secondly, we advocate for complete overturn of the slab within the mantle transition zone. Figures 2d-2f indicate that the 2015 Bonin Islands earthquake ruptured the lower edge of the imaged high velocity anomaly. Assuming the highest velocities correspond to the coldest parts of the slab, and using thermal modelling results which indicate the coldest parts of the Izu-Bonin slab are about 30 km from the slab surface (e.g. Emmerson and McKenzie, 2007), the earthquake occurred <25 km from the crustal section of the slab. This interpretation differs from other analyses of the 2015 Bonin earthquake, which locate the earthquake close to the “bottom” of the subducting slab about 100 km from the crustal section, and therefore in a high temperature region (Porritt and Yoshioka, 2016; Takemura et al, 2016; Obayashi et al, 2017; Zhao et al, 2017). As all potential mechanisms for deep-focus earthquake generation require relatively low temperatures (Frohlich, 2006), our new

interpretation resolves what was a major problem for theories of deep earthquake generation. Nevertheless, our results indicate that the 2015 Bonin Islands earthquake joins a growing list of large, isolated deep-focus earthquakes which are believed to lie close to the edges of subducting slabs: Spain 1954; Peru, 1963; Colombia, 1970; Bolivia, 1994; Tonga, 1994 (Green and Houston, 1995); and more recently Okhotsk, 2013 (Zhan et al, 2014). For this reason, the 2015 Bonin Islands Earthquake may have ruptured a region experiencing high thermal gradients and more rapid heating than most of the slab.”

We also explicitly include 2015 Bonin locations from both our and Zhao et al. (2017) studies, as follows:

“Our DD tomographic inversion also simultaneously relocates the 2015 Bonin earthquake at latitude 27.741°N, longitude 140.572 °E, and depth 679.9 km, which is deeper than the location (latitude 27.740°N, longitude 140.590 °E, and depth 667.2 km) given by Zhao et al. (2017) but with a similar epicenter. The restoration test shows that our location uncertainty in depth is about 4.5 km.”

Line 241: I disagree. Based on the data and synthetic tests shown I am not convinced that e.g. tears can be well resolved. I suggest to show end member tests to prove this statement.

Reply: We have conducted more synthetic tests (including end member models) to show the main slab features that we discuss are well resolved. Please see Supplementary Information for details.

Line 242: better: 'relative slab thickening' as you write above that the absolute thickness cannot be estimated.

Reply: Agreed, modified as suggested

Line 243: 'tests' is a strange word when speaking about real data. What about "reference dataset" or something similar?

Reply: Agreed. We have changed “tests” to “reference data” as suggested.

Line 244: “vp images show”  better: 'suggests' as it is not an observation but your interpretation

Reply: Changed as suggested.

Line 245: For me it is still not clear: what exactly are the authors' arguments for non-linear rheology?

Reply: We added a paragraph earlier in the text (see above) that has made our arguments clearer. In essence, any shear localization/"tearing" requires a non-linear rheology. Our new tomographic images clearly imply shear localisation.

Reply: Line 248: why do you not write here at which location you obtain the earthquake hypocenter? Isn't this one of the main findings?

Reply: We have included the location information in the Results section, as follows:

"Our DD tomographic inversion also simultaneously relocates the 2015 Bonin earthquake at latitude 27.741°N, longitude 140.572 °E, and depth 679.9 km, which is deeper than the location (latitude 27.740°N, longitude 140.590 °E, and depth 667.2 km) given by Zhao et al. (2017) but with a similar epicenter. The restoration test shows that our location uncertainty in depth is about 4.5 km."

Line 248-251: I am not sure how these two last sentences should add any information to the conclusion (except advertising another paper by the first author)? -- Sorry -- I suggest to remove these sentences

Reply: We have deleted the two last sentences as suggested.

Figures:

Figure 1: Possibly scale the earthquakes by magnitude and specify in the caption which time of EHB data is plotted.

Reply: The range of years for the EHB catalogue locations (1960-2008) are added to the text. We have chosen not to scale the earthquakes, as the resulting image would look too cluttered.

Figure 2: Why is figure a-c cut to the west? It would be nice to see how the slab continues to the west (until the resolution limit is reached)

Reply: We have now updated Figures 1–3 to show the slab another 200 km further west.

What do you mean with "representative earthquakes"? It would be more useful to plot e.g. the largest earthquakes, possibly also highlighting their aftershock sequences.

Reply: In Figure 2, we are interested in the potential link between fault planes and shape of velocity anomalies in our models. There aren't many fault identifications in the literature; we here plot those from Myhill and Warren (2012), and also the 2015 Bonin event mechanism from Ye et al. (2016). The uneven distribution of seismicity means that including all the events from Myhill and Warren would just add clutter, so we pick a representative selection. The caption now reads:

“Tomographic images of P-wave velocity and relocated earthquakes within the sections shown in Figure 1. Note that in Sections d-f the high velocity anomaly appears to split in the mantle transition zone, This is partially a consequence of limited resolution (especially in f), but also due to the orientation of the slab tear; Sections d and e show parts of the slab on both sides of the inferred slab tear (see Figures 1 and 3). Also shown are focal mechanisms and rupture planes for moderate to large earthquakes ($M_w > 5.7$) in the mantle transition zone determined by directivity analysis in a previous study (Myhill and Warren, 2012), rotated into the plane of section. The EXX codes represent the catalogue IDs in that study. The rupture planes for each earthquake (Myhill and Warren, 2012; Ye et al., 2016) are highlighted in blue.”

It is unclear to me how the resolution limit is determined

Reply: We have now described how we determine the resolution limit in the supplementary information, as follows:

“Figures S5 and S6 show recovered checkerboard patterns for the coarser inversion grid at different depths and latitudes. Figures S7 and S8 show recovered checkerboard patterns for the finer inversion grid. The checkerboard tests suggest that the inverted model is well-resolved horizontally and vertically and that the high-velocity structure of slab in the Izu-Bonin area is well-defined below 100 km for the selected inversion grid. We also tried using finer inversion grid nodes in latitude with an interval of 1° from $8^\circ N$ to $36^\circ N$. As shown by the checkerboard test results, the resolution becomes degraded compared to the inversion grid used for real data inversion (Figure S9). For this reason, we think the currently selected inversion grid is a good compromise between resolution and robust retrieval of velocity anomalies.”

In the text you refer to the sub-figures with capital letters; in the figure they have small letters - please unify this.

Reply: The manuscript text has been changed to have small letters.

Figure 3: ‘the shear zone’  ‘the inferred shear zone’

Changed.

Why is the overturned part blue? Should this be a shadow?

The overturned part is blue only to highlight it. The caption now reads:

“The blue region marks where the slab is inverted.”

Please write at some point that the green area is the 660.

The caption now reads:

“The dark gray surface marks the “660”-km depth isocontour.”

Supporting information:

Line 41: How are the outliers removed? Visually? Based on some mathematical criterion?

Reply: The outliers are removed visually based on the main trend of travel time curves. The sentence is revised as follows:

“After removing the outliers outside the main trend of travel time curves in the EHB catalog, 893,359 absolute P wave arrival times are selected for the 9998 earthquakes in the Izu-Bonin region (Figure 1) recorded by global stations in the period of 1960 to 2008.”

Line 43: Is the maximum epicentral distance 100 degree? Is there any special treatment for Pdiff?

Reply: Yes. As shown in Figure S2, the maximum epicentral distance is 100 degrees. For this reason, our arrival time dataset does not include Pdiff.

Line 47: an average offset of 54 km seems quite a lot to me. What are the maximum/minimum values? Did you put any constraints on these? Did you do tests with smaller/larger offsets?

Reply: We have revised the text to address this concern as follows:

“We require the minimum offset for event pair is 2 km when constructing differential arrival times. For the assembled differential arrival times, the average offset for all the event pairs is 54.7 km and the maximum offset is 299.97 km. Generally the average offset should be close to and greater than the minimum inversion grid interval and in this case it is 30 km in the longitude.”

For this study, we did not try constructing another set of differential arrival times, as we think the current dataset is suitable for imaging main slab features of interest.

Line 169: ‘180km’  ‘180 km’

Changed.

Lines 94-95: Isn’t this conclusion on the resolution limit a little bit over-simplified? The resolution also varies laterally. It is still not clear to me based on what information the very complex resolution limit in Figure 2 is constrained.

Reply: In the revised supplementary materials, we have added more synthetic tests and more figures to illustrate that the assembled dataset combined with teletomoDD has the ability to resolve the main slab features that we inverted.

Line 105: I am quite surprised to see so much negative energy in the recovered velocity model. Where does it come from? Is the corresponding energy mapped in station or event terms?

Reply: Thanks very much for pointing this fact. We were also puzzled when carefully looking at negative anomalies appearing below the slab because the checkerboard tests indicate this region is well resolved. It turns out for this synthetic test the velocity values for the grid nodes at one depth below the 660 km were mistakenly set with lower values. Once this mistake is corrected, the synthetic tests no longer show these low velocity artifacts in the recovered models.

Line 179: Full stop missing.

Added.

Figure S1: Please add ellipses to indicate epicentral distance from study region.

Reply: We have added ellipses of 50° and 100° to indicate epicentral distance from study region.

Figure S4: This is an irritating representation for the checkerboard tests – please add an explanation in Figure caption: are the rectangles the values at each node? Possibly show in the same interpolation than the real data? Use same color scheme and model geometry representation than for S5 and S6.

Reply: Following your suggestion, we have used another way to represent checkerboard models.

Figure S6: Please add the geometry of the input anomaly on top – otherwise it is very hard to see differences. Possibly add the hypocenters prior to inversion to see their shift. Use exactly the same color scheme than in S5.

Reply: In all the synthetic tests, we have added the hypocenters in the “true” and recovered models to indicate likely shift for the anomaly.

Figure S5: I think it is irritating to choose such a complex input synthetic model. It is hard to see which features are recovered, smeared etc.. I suggest to design an easier model (e.g. cubic forms with constant amplitudes) which has the characteristics of the real data.

Reply: Following your suggestion, we have included more and simpler synthetic models to show the interpreted features can be well resolved.

Figures S5/S6: Where are Figs. S5 and S6 located on the map? The same nomenclature as in Fig. S4 should be used.

Reply: For the cross sections shown for all synthetic tests, the profile locations are the same as those shown in Figure 2. We have made this clear in the figure captions.

Figures S5/S6: In the main text you write that you estimate the resolution to be ~200 km in N-S? Where does this information come from? Please plot horizontal sections here as well - it's hard to understand otherwise.

Reply: We have included horizontal sections of recovered checkerboard models in the supplementary materials. With the refined inversion grid, the model resolution can be up to ~100 km in N-S.

Figure S6 exists twice!

Changed.

Figure S7: Label a,b,c are missing. The annotation of the y axes is missing.

Reply: This figure has been modified to make the text larger, and the a, b, and c labels have been added. The x axes and y axes are the same for all three plots, so the label is only included once.

Reviewer #2 (Remarks to the Author):

In this manuscript, the authors present tomographic images of the subducting slab in the Izu-Bonin subduction zone and evaluate the relationship of the imaged structure to the 30 May 2015 deep earthquake. This earthquake is extremely unusual due to its depth, magnitude, and distance from the Wadati-Benioff zone and therefore understanding the processes leading to this earthquake provide rare insights into deformation near the base of the mantle transition zone.

Reply: We really appreciate your opinion on the importance of our study and your comments on our paper that help improve its quality.

Their model is significantly higher resolution than previous models due to their double difference methodology, which has maximum resolution in regions of high seismic activity. I would appreciate a discussion (briefly in the main text and slightly less brief in the supplements) concerning the weaknesses in the method. For instance, I would assume that due to the double differencing, the results can be more skewed than traditional body wave tomography by outliers. This is why careful rejection of outliers (e.g. Fig. S2) is so important.

Reply: In the supplementary materials, we have included more explanations on possible weakness of the method, as follows:

“Due to differencing in travel times for pairs of events on common stations, differential arrival times may contain larger random errors. Conversely, the differencing operation can greatly reduce the effect of systematic errors in absolute arrival times (Zhang and Thurber, 2003). Waveform cross correlation can be used to determine relative arrival times among events much more accurately than absolute times (Waldhauser and Ellsworth, 2000), as long as the two events are close together and have similar focal mechanisms. Finally, in the double-difference tomographic method, absolute arrival times are used in addition to double difference measurements, in order to improve absolute event locations (Zhang and Thurber, 2003).”

In the main text, we add the following sentences concerning the weakness in the method:

“If differential arrival times for event pairs are not constructed by the waveform cross correlation technique, however, they may contain larger random errors due to differencing in absolute arrival times. Thus outliers in arrival time picks should be carefully rejected.”

I would also like to know why the spatial resolution in latitude is an order of magnitude worse than in depth and longitude. This seems odd and comes back in line 153 when they discuss that the low velocity anomaly is no more than ~50 km wide and it is unclear in which direction this refers.

Reply: During the revision process, we have carefully considered concerns from both reviewers on the model spatial resolution in latitude. We now take a two-step approach by using a coarser inversion grid first and then a finer inversion grid. Now the current inversion grid has the spatial resolution of 1° around the 2015 Bonin earthquake and 2° for the region away the 2015 Bonin earthquake. The current inversion grid nodes still are coarser in latitude than in longitude and depth. This is because our main purpose is to see the slab pattern in the cross section normal to the trench, which is roughly parallel to longitude.

For the low velocity anomaly of ~50 km width, we are referring to the distance along the slab dip direction. We have now made this clearer.

This also leads me to ask how the mesh inversion mesh was defined. Did the authors simply say that this is the region of interest and has high seismicity or did they use a data adaptive meshing algorithm (this is probably a supplementary information topic)?

Reply: The way we select inversion mesh or grid is through a hierarchical process. We first select coarser inversion grid and check if the model can be well resolved with it. If so, we then refine the inversion grid to further check the resolution of the model. The general way to check model resolution is the checkerboard test. We have added more details on how to set the inversion grid used in this study in the supplementary materials.

The primary findings of the model indicate 4 major regions. The northernmost region is a

steeply dipping slab, which flattens in the transition zone. South of that, the slab has the same geometry, but a horizontal tear is developing which is separating the steeply dipping slab from the slab lying flat in the transition zone. Further south, between 26.5° and 28.5°, is a large-scale plunging tear. The southernmost section, south of the tear is where the slab is overturned in the lower mantle with the top of the slab lying on the base of the transition zone. It is in this, southernmost overturned part of the slab, that the earthquake occurred. Due to the apparently relatively low resolution in latitude, it is unclear how well separated these regions are. A resolution map or north-south cross section may clarify this uncertainty.

Reply: This is really an excellent summary of our major findings! We have included horizontal sections of recovered checkerboard models to show spatial resolution in latitude in the supplementary materials.

Considering the scatter in ISC data, I'm curious how much the structure is dependent upon that data. There are two types of tests that could be utilized for this. First, rerun the inversions using other publicly available datasets. The first dataset that I can think of is the Array Network Facility, which provides USArray delay times. That will likely have bias due to the limited aperture of the array but could be combined with other datasets. An alternative would be to bootstrap resample the travel times and see how much that causes the structure to vary.

Reply: We have conducted bootstrap analysis by randomly leaving out 10% of data each time. In total, 50 times of inversion is performed. The standard deviation for the regional model is lower than 0.06% and most the model region above 660 km has the value lower than 0.04% (Figure S19). This indicates the model structure is robust and stable.

There is something unclear in regards to the geometry of the slab as illustrated in Figure 3 and Figure 2e,f. The horizontal slab in the transition zone could be inferred to lying both east and west. This is similar to the uncertainty in geometry shown by Obayashi et al., 2013 and is dependent upon your resolution in latitude.

Reply: In our interpretation, the slab in sections 2d and 2e is indeed lying to both east and west. This is geometrically possible because of the orientation of the slab tear. We interpret the muted high velocity anomaly in Figure 2f as the result of limited resolution in latitude, rather than the presence of a slab in the region.

Figure 3 has been slightly modified to better highlight the orientation of the slab tear. An extra sentence has been added to the caption of Figure 2 describing in more detail our interpretation of the tomographic images.

Line 181: add parentheses around -0.4 to separate it from the ~ symbol.

Reply: We've replaced the tilde with the word approximately in this case, to avoid confusion.

The supplementary information could use careful proofreading. It is mostly good, but there are a few confusing sentences.

Reply: We have read more carefully the supplementary information and corrected some grammatical errors.

I would also ask the authors, if possible, that they make the 3D Vp model publicly available as research into this earthquake will likely be ongoing.

Reply: Yes, we will make the model available in the supplementary materials or through an open-access website.

Overall, I find this to be a good paper, requiring only minor revisions. I don't see this as inherently contradictory to my earlier work on the region, but rather I think this higher resolution model is consistent with the data I presented and sheds valuable light onto the ongoing processes of the region.

Rob Porritt

References:

Obayashi, M., J. Yoshimitsu, G. Nolet, Y. Fukao, H. Shiobara, H. Sugioka, H. Miyamachi, and Y. Gao (2013), Finite frequency whole mantle P wave tomography: Improvement of subducted slab images, *Geophys. Res. Lett.*, **40, 5652–5657, doi:10.1002/2013GL057401.**

Reply: Thanks very much for your comments. We have added a sentence to the discussion:

“Our interpretation is also largely consistent with the data of Porritt and Yoshioka (2016), whose P to S receiver function data indicated that seismic energy passing upward through the source region of the 2015 earthquake traversed more than one region of high seismic velocities.”

Reviewers' comments:

Reviewer #1 (Remarks to the Author):

I am happy to hear that you found most of my comments useful. I think that the resubmitted paper reads very well and is logically easy to follow. Thanks for taking the time and effort of conducting further tests on the model resolution. However, to be very honest, I have to admit that I consider some of the comments I had to point out in the first review not as the actual task of the reviewer, when submitting to a Nature-family journal. E.g. inconsistencies in figure labels, missing references etc. should ideally be found by the group of co-authors within the internal review process, prior to submitting the paper.

I have only one more general and some minor comments.

General Comment: In figure 2, there appears a high velocity zone at kilometer 700 to 850 and at depths from 0 to 400 km. This feature seems separated from the slab you are discussing and dipping with different angle. Could you please comment on the origin and the possible interpretation of this anomaly?

Minor Comments (please note that the line numbers given here refer to the article file in which changes are highlighted):

Line 35: 'a tear that dips ..'  Actually a tear cannot dip; or it is not totally clear that you are referring to the slab dip here. Possibly slightly reformulate this sentence. E.g. something like 'a tear in the northward dipping slab'?

Line 153 to 154: I think this is a bit too much. Different researchers will have quite different opinions which is the 'most enigmatic earthquake ever recorded'. Could you please write something like 'a very enigmatic deep earthquake' or 'to our opinion..'

Line 248: 'Figure 2F'  'Figure 2f'

Line 253-254: 'In Sections AA'-CC', the locations of the high velocity anomalies are in excellent agreement with previous interpretations based on seismicity and tomography.'  is there a reason for this good agreement in one part of the study region and the worse agreement in other parts of the study region? E.g. 'because sees studies were using local networks in this region' .. or something similar?

Line 656: 'The blue region marks where the slab is inverted'  'The blue region marks where the overturned slab is inverted'

Minor Comments Supplement (please note that the line numbers given here refer to the updated file without changes highlighted):

Line 124 to 125 (and the entire paragraph): I think you should not write 'low velocity shear zone' here, as your synthetic model does not have low, but only neutral velocities there. You could write 'not elevated with respect to the background velocity model'

Line 130 (and 131): 'The model shown in Figure S15 shows that slab' (and the following sentence with similar structure)  possibly use the word 'show' only once.

Line 132-134: This sentence is hard to understand and I am not sure if the grammar is totally correct. Possibly you could split this sentence into two and skip at least one 'also'.

Reviewer #2 (Remarks to the Author):

The authors have adequately responded to my earlier concerns and those of the other reviewer. I have no further comments at this point.
-Rob Porritt

Thanks again for the constructive and thoughtful reviews from reviewers. We have marked the original comments in **bold**, our replies in regular, and newly added text in the revised paper in *italic* fonts, respectively.

Reviewers' comments:

Reviewer #1 (Remarks to the Author):

I am happy to hear that you found most of my comments useful. I think that the resubmitted paper reads very well and is logically easy to follow. Thanks for taking the time and effort of conducting further tests on the model resolution. However, to be very honest, I have to admit that I consider some of the comments I had to point out in the first review not as the actually task of the reviewer, when submitting to a Nature-family journal. E.g. inconsistencies in figure labels, missing references etc. should ideally be found by the group of co-authors within the internal review process, prior to submitting the paper.

Reply: Thanks again for your thorough and thoughtful reviews. It is great to know that the revised version addressed your previous concerns. We have carefully checked the current version to make sure there are no apparent word, figure and reference issues. We hope that you are satisfied with the new version.

I have only one more general and some minor comments.

General Comment: In figure 2, there appears a high velocity zone at kilometer 700 to 850 and at depths from 0 to 400 km. This feature seems separated from the slab you are discussing and dipping with different angle. Could you please comment on the origin and the possible interpretation of this anomaly?

Reply: In the revised Discussion section, we briefly discussed these two features, as follows:

“Although the slab appears to be stagnant in the transition zone in the study region, it is possible that a small fraction of the slab in the north of the study region may penetrate through the 660 km interface, as indicated by relatively well resolved high velocity anomalies below 660 km in sections AA’ and BB’ (Figure 2). To the east of the main Wadati-Benioff zone, there exists a high velocity zone at depths from 0 to 400 km that dips away from the slab. The resolution tests (see Supplementary Information) show that at depths from 0 to 200-400 km around the eastern edge of the study region where the high velocity anomaly is located, the model is not well resolved due to poor angular coverage of ray paths. Therefore, it is likely the high velocity anomaly at the edge of the model is not robust and no further interpretation is attempted here. ”

As discussed above, we think high velocity anomalies at kilometer 700 to 850 likely to be the penetrated part of the slab. But as seen from Figure 2, a great majority of the slab is stagnant in

the mantle transition zone. The high velocity zone at depths from 0 to 400 km is at the edge of a region of poor model resolution, and may not be robust. We are therefore reluctant to interpret it. We have added new figures S10 and S11 to show semblance or model resolvability distributions along 6 profiles for both models with coarser and finer inversion grids to indicate that the high velocity zone around eastern edge of the model is likely poorly resolved.

Figure 2 and Figure S4 are slightly updated to show poorly resolved model regions with less saturated colors (or shaded regions) based on model resolvability values for coarser and finer inversion grids. In the Supplementary Information, we give more details on how to mask poorly resolved model regions based on checkerboard resolution test results for coarser and finer inversion grids. The details are as follows:

“To quantitatively assess the resolvability of checkerboard models, we calculate the semblance value at each grid node between true and recovered checkerboard models using nearest nodes in three directions following the way of Guo et al. (2018) which is modified from the method of Zelt (1998). We calculated the model resolvability for both coarser and finer grids. Figures S10 and S11 show the distribution of resolvability values along the same 6 profiles as those in Figure 2 for coarser and finer grids, respectively. It can be seen from Figures S10 and S11 that the model resolvability is high around and below the slab. However, in the shallower part above the slab and around the left and right edges of the model cross-sections, the model resolution is relatively poor with resolvability values < 0.7. These resolvability values are then used to determine the well-resolved model regions. For the coarser grid, the model region is treated as well resolved if the resolvability value at each grid node is larger than 0.8 (Figure S4). For the model inverted using the finer inversion grid, we combine resolvability values obtained for coarser and finer grids to determine the well-resolved model region. This is because the model using finer inversion grid is inverted using coarser inversion grid model as the starting model so that any artifact from the coarser grid model would be brought onto the finer grid model. The model at finer grid node is treated as well resolved if the resolvability with the finer inversion grid is larger than 0.75 and the resolvability with the coarser inversion grid is larger than 0.8 (Figure 2).”

Minor Comments (please note that the line numbers given here refer to the article file in which changes are highlighted):

Line 35: ‘a tear that dips ..’  Actually a tear cannot dip; or it is not totally clear that you are referring to the slab dip here. Possibly slightly reformulate this sentence. E.g. something like ‘a tear in the northward dipping slab’?

Reply: This sentence is now revised as follows:

“We resolve a tear in the slab which can be approximated by a northward-dipping plane that splits the slab in the mantle transition zone between 26.5° N and 28° N (500 to 670 km depth).”

Line 153 to 154: I think this is a bit too much. Different researchers will have quite

different opinions which is the ‘most enigmatic earthquake ever recorded’. Could you please write something like ‘a very enigmatic deep earthquake’ or ‘to our opinion.’

Reply: We have revised it as suggested:

“High-resolution seismic tomography could also help resolve the mystery of a very enigmatic deep earthquake.”

Line 248: ‘Figure 2F’  ‘Figure 2f’

Reply: Changed.

Line 253-254: ‘In Sections AA’-CC’, the locations of the high velocity anomalies are in excellent agreement with previous interpretations based on seismicity and tomography.’ -> is there a reason for this good agreement in one part of the study region and the worse agreement in other parts of the study region? E.g. ‘because sees studies were using local networks in this region’, or something similar?

Reply: Thanks for this comment. We have added one sentence to explain this agreement, as follows:

“This good agreement is probably due to the relatively simple shape of the slab in the north of the study region, and the proximity to the dense Japan seismic network.”

Line 656: ‘The blue region marks where the slab is inverted’  ‘The blue region marks where the overturned slab is inverted’

Reply: The sentence has been rephrased as follows:

“The blue region marks where the slab is overturned.”

Minor Comments Supplement (please note that the line numbers given here refer to the updated file without changes highlighted):

Line 124 to 125 (and the entire paragraph): I think you should not write ‘low velocity shear zone’ here, as your synthetic model does not have low, but only neutral velocities there. You could write ‘not elevated with respect to the background velocity model’

Reply: We have revised this sentence to make the point clearer, as follows:

“To further test the robustness of the low velocity anomaly around 400 km in depth with respect to the shallower and deeper parts of the slab appearing in the inverted model (Figure 2), we constructed a synthetic slab model similar to that in Figure S12 but containing a low velocity zone around depth 400 km (Figure S14). ”

Line 130 (and 131): ‘The model shown in Figure S15 shows that slab’ (and the following sentence with similar structure)  possibly use the word ‘show’ only once.

Reply: Corrected. We have deleted “shown” in both sentences.

“The model in Figure S17 shows that slab overturn cannot be retrieved as an inversion artifact. The model in Figure S19 shows that the inversion procedure can also recover the case where the slab overturns throughout the area.”

Line 132-134: This sentence is hard to understand and I am not sure if the grammar is totally correct. Possibly you could split this sentence into two and skip at least one ‘also’.

Reply: We have split this sentence into two sentences, as follows:

“In the slab model shown in Figure S16, we include the (relatively low) velocity anomaly in the slab around depth 400 km. This feature is also well resolved (Figure S17).”

Reviewer #2 (Remarks to the Author):

The authors have adequately responded to my earlier concerns and those of the other reviewer. I have no further comments at this point.

-Rob Porritt

Reply: We are happy to know that you are satisfied with our revised version. Thanks again for your comments that were very helpful for improving this paper.

Reviewer #1 (Remarks to the Author):

The authors have addressed all my concerns adequately and I do not have any further comments.
Good luck with your work!